# Esthetic Rehabilitation of Pediatric Patients Using Direct Bonding Technique—A Case Series Report

**DOI:** 10.3390/children10030546

**Published:** 2023-03-13

**Authors:** Sittana Elfadil, Hossam I. Nassar, Rana Aly Elbeshbeishy, Lovely M. Annamma

**Affiliations:** 1Department of Clinical Sciences, College of Dentistry, Ajman University, Ajman P.O. Box 346, United Arab Emirates; 2Center of Medical and Bio-Allied Health Sciences Research, Ajman University, Ajman P.O. Box 346, United Arab Emirates; 3Department of Prosthodontics, Future University Egypt, New Cairo P.O. Box 74, Egypt; 4Visiting Faculty, Department of Basic Sciences, College of Dentistry, Ajman University, Ajman P.O. Box 346, United Arab Emirates; 5Visiting Faculty, Department of Clinical Sciences, College of Dentistry, Ajman University, Ajman P.O. Box 346, United Arab Emirates

**Keywords:** pediatric esthetic dentistry, amelogenesis imperfecta, direct veneers, composite resin, dental spacing, microdontia

## Abstract

Pediatric esthetic dentistry is a sensitive technique, as children can be less cooperative; hence, minimal appointments are preferred. The most conservative treatment modality for anterior esthetic rehabilitation is using direct composite veneer restorations. In many instances, esthetic improvements using composite resin are the only possible option until the growth phase of the pediatric patient is complete. In this article, we present three different case scenarios in young teenagers with different treatment needs requiring esthetic restorations. The first case report is a patient with amelogenesis imperfecta, the second is with generalized spacing, and the third is with localized microdontia. All three patients were followed up for six months post direct composite veneering and were highly satisfied with their treatment outcomes, which boosted their self-confidence.

## 1. Introduction

In pediatric dentistry, many materials such as glass ionomers, resin-modified glass ionomers, compomers, and composite resin are employed as restorative materials. Of all the materials, composite resins are the most esthetic and desired for anterior restorations [1]. Using direct composites as veneering materials for pediatric patients offers the convenience of treatment completion in a single appointment and has the added benefit of the possibility of replication or changing of existing tooth morphology, proportion, and color [2]. The need for esthetically restoring anterior permanent teeth in young children is commonly performed for fractured, discolored, or anterior teeth with spacing. Discolored teeth in children needing restorations can be due to caries, fluorosis, intrinsic discoloration, and congenital defects such as enamel hypoplasia and amelogenesis imperfecta [3].

Children can be highly conscious of their esthetics, as per the study of vale et al. [4,5]. The lack of esthetics can bring down the confidence and morale of a school-going child and can affect academic performance, too. Several authors have successfully restored anterior dentition in pediatric patients with composite resins and have followed up on their cases to evaluate the restorations’ success [6,7]. Techniques in which composite resin can be utilized on anterior teeth for restoration include direct technique, indirect technique, and semidirect technique [7].

In this case series, three different pediatric patients with varying esthetic complaints were selected. All patients and their guardians were informed of the advantages, disadvantages, materials, and techniques of each treatment option. The patients opted for the direct veneering technique with composite due to its simplicity and, subsequently, informed consent was obtained.

## 2. Case Series

### 2.1. Case Report One: Amelogenesis Imperfecta

A 13 year old female patient previously diagnosed with amelogenesis imperfecta reported to the dental clinic seeking an esthetic correction. She was medically fit and had regularly attended review visits with the state pediatric dentist in the past. On clinical evaluation, she had a fair oral hygiene status, caries-free dentition, preventive fissure sealant restorations on posterior dentition, and old stained partial composite veneers on maxillary incisor teeth. Localized marginal gingivitis was present around the anterior teeth due to plaque retention on rough teeth surfaces. The patient underwent two years of fixed orthodontic treatment to align her teeth. Effective bonding was maintained throughout orthodontic treatment, confirming the good quality of existing enamel. After orthodontic debonding, the patient complained that her teeth were “dark” and “stained”. Her teeth were stained with pitted hypoplastic enamel that varied in color from yellow to deep brown discoloration (Figure 1a,b and Figure 2 Case 1)

#### Clinical Procedure

Based on the esthetic analysis, the maxillary incisor teeth were found to be oval in form due to cervically positioned contact points and deep interdental embrasures with anterior spaces, despite recent orthodontic alignment. The incisal edges of the maxillary anterior teeth had a straighter alignment that did not follow lower lip curvature while smiling.

Due to her young age, incomplete maturation of gingival positions along with residual altered passive eruption was evident; hence, the use of indirect fixed restorations was not indicated. Direct layered esthetic composite restorations were offered as a medium-term solution to address her esthetic concerns until the patient passes her growth spurt. In the long term, this treatment was to be replaced by ceramic veneer. Complete esthetic and smile analyses were completed based on various records including repose and smile photographs along with teeth and space measurements utilizing the esthetic guide by S.J Chu et al. [8]. Mock-up was used to highlight the proposed form of restorations, test phonetics, and verify all aspects of esthetic designing including incisal display, smile form, and teeth proportions. Both the patient and father were very pleased and approved the mock-up. A palatal putty index was generated for future reference. Consent was gained for eight direct layered composite veneers between the right and left maxillary first premolars.

The patient underwent periodontal debridement and completed a course of non-vital home bleaching using 16% carbamide peroxide (Opalescence PF 16%; Ultradent, South Jordan) using her Essix retainers as bleach trays to improve the initial shades of the teeth. Two weeks were allowed before the final shade selection. The clinical procedure for direct veneers was conducted under a rubber dam isolation using floss ligation (Figure 2). The defective and stained old composite veneers were removed and the remaining stained enamel areas were polished.

The teeth were sandblasted using an intraoral sandblaster, the enamel was etched with 37% phosphoric acid, adhesive bonding was applied (Scotchbond Universal, 3M ESPE), and was followed by a layered composite technique with a nanohybrid composite resin material (Inspiro^®^, EdelweissDR, Germany). Esthetic layering techniques were used utilizing putty palatal index and composite with different shades, tints, and translucencies (Enamel/Skin and Dentine/Body composites). Diamond finishing burs and Soflex discs were used for finishing to produce macro- and micro-anatomy. Polishing wheels were used (Diatech Shape Guard Polishers, COLTENE) to produce life-like high-luster restorations. Occlusion was adjusted and canine guidance was maintained. An alginate impression was made immediately post-bonding to fabricate a new maxillary Essix retainer to prevent relapse and protect composite restorations. The patient was reviewed after a few months, and she was very pleased with her esthetic outcomes (Figure 3).

### 2.2. Case Report Two: Anterior Dentition Spacing

A 16 year old teenager reported to the clinic complaining of a lack of confidence and was anxious to smile due to spaces in between her teeth. She was medically fit, and clinical examination revealed good oral hygiene status with chronic mild gingivitis (Figure 4).

The patient had a well-maintained dentition with only two small Class I amalgam restorations on maxillary first molars. Teeth and space measurements along with smile analysis and designing were done. The following observations were recorded: medium smile line, worn incisal edges causing teeth shortening, and narrow teeth with spaces between all anterior teeth. The teeth shape was square in form and the incisal plane was not parallel to the lower lip line (Figure 5).

Multidisciplinary orthodontic-restorative treatment planning was completed, aiming to consolidate the spaces on the mandibular arch and favorably re-distribute maxillary anterior spaces for future esthetic enhancement. Accordingly, prosthodontic planning and prescription of the final teeth positions were given to the orthodontist. After orthodontic alignment, the smile design was redone (Figure 6).

The same technique as for case one was performed to close the remaining spaces esthetically with Inspiro composite resin (Figure 7). Special consideration was taken while building proximal composite additions to produce a proper emergence profile, generate correct incisal embrasure ratios, and accurately position the height of the proximal contacts. Post-operative papillary fill was evident since proximal contacts were placed following the recommendations of Tarnaw et al. [9], where almost 100% of the time papilla is expected to fill when the contact point is 5 mm or less away from the alveolar bone. The patient was highly satisfied with the final smile (Figure 7). Follow-up was conducted after 6 months, where esthetic outcomes remained the same and the patient reported no complications with the restorations.

### 2.3. Case Report Three: Localised Microdontia

A healthy 13-year-old patient underwent orthodontic treatment to optimize anterior spacing and improve teeth alignment. Just before debonding, the orthodontist referred the patient to the restorative dentist for treatment planning and smile analysis. The patient complained of gaps between his narrow teeth (Figure 8).

Clinical examination showed fair oral hygiene, localized anterior chronic gingival inflammation, and localized microdontia affecting the maxillary incisor teeth. The four maxillary incisors’ proportions were distorted with adequate crown heights but were reduced in width. Maxillary lateral incisors were peg-shaped with further distortion in shape, form, and proportion. Smile analysis confirmed the correct positions of teeth and the adequately sized proximal contact needed for esthetic rehabilitation. Therefore, no further teeth movements were needed and orthodontic debonding was planned.

The clinical procedure of composite layering was performed using a palatal putty index generated based on an approved mock-up. This was essential to accurately and predictably establish the correct form and proportion of the maxillary peg lateral incisors (Figure 9).

Esthetic goals were achieved, and the patient was pleased with the final smile improvement (Figure 10). On review, the patient had no concerns, but some restorations required further polishing. The patient was regularly wearing a hard Essix retainer every night.

## 3. Discussion

The direct veneering technique is extremely successful for pediatric cases as it is economical, extremely conservative, and can be completed in minimal appointments while providing immediate results. It is most advantageous in pediatric dentistry, where fixed direct restoration has limited indications among growing patients with immature gingiva and large pulps limiting treatment plan options. The use of composite resin for esthetics in children is highly recommended due to its reversibility, versatility, simplicity, and its potential to be upgraded past the growing stages to fixed treatment options. Alterations in composites can be performed in the color, form, shape, and proportion of teeth to a limit.

Amelogenesis Imperfecta involving enamel deformation, if treated with ceramic crowns, can possess a high risk of pulpal exposure in children. When good enamel bonding qualities are evident, such as in our first case report, the preferred conservative treatment would be composite restoration. Many authors have reported success with various composite resin techniques for amelogenesis imperfecta patients [7,10,11]. The esthetic correction for generalized spacing in children and peg lateral shaping with direct veneered composites are also documented in the literature [12].

Direct layered composite veneers were the mode of treatment selected by all three presented patients since they produced an excellent esthetic solution and successfully acted as a transitional treatment modality during the important high school and college years of pediatric patients’ lives where ceramic veneers are contraindicated.

Many authors have proved that composites can be used as a long-term restoration for esthetic correction, provided the patient follows routine oral hygiene measures [13,14], with an observed longevity of 80–89% after 5 years [12,15]. Common failures include composite resin chipping and fracture and the need for polishing from time to time. The direct composite techniques are mostly additive in nature, and therefore would not be ideal in cases of severe enamel deficiency and clinical scenarios requiring subtractive restorative techniques. Unfortunately, composite veneers are quite a sensitive technique, requiring a highly skilled operator and cooperative patient. Composite veneers are recommended as the most conservative treatment option in pediatric patients with discolored or traumatic teeth, or with teeth with hypoplastic changes. The direct bonding technique has many advantages compared with the few disadvantages, such as the need for proper isolation, skill, and additional chairside time [16].

The longevity of composite as an anterior restorative material was evaluated by Comba et al. in a five-year retrospective study on post-orthodontic patients. They concluded that even after 5 years composite, when used as an anterior esthetic corrective material, had satisfactory results. The few cases that failed were repairable and, hence, composites are the most conservative and ideal treatment for teenagers and growing children [17]. The advantage of using the direct bonding technique is that it can be polished and finished adequately to prevent any gingival irritation. The contraindications mentioned in the literature include cases where isolation cannot be achieved, with caries-prone teeth, and lack of patient follow-up [18]. Recen et al. compared 15 cases, each performed by indirect composite and direct composite techniques. They concluded that the direct technique is significantly better than the indirect composite veneer technique. According to their findings, the indirect composite veneers were more prone to discoloration and required complex preparation such as an impression technique, as well as lab procedures [19]. In another study, the type of composite used for the direct technique was evaluated by Mewan; he concluded that the nanohybrid composite used with an indirect–direct technique had maximum longevity compared with micro-filled composites. In both direct and indirect techniques the nanohybrid composites had decreased gingival microleakage compared with the micro-filled composites [20].

## 4. Conclusions

Layered direct composite restorations are excellent, well-documented, and widely used treatment options among adults. It is specifically and especially more convenient and advantageous to be used among teenagers and growing children. These patients have very limited treatment options to improve their dental esthetics, while it is highly required to avoid peers’ pressure during critical teenage life. Case selection, patient cooperation, operator skill, good communication with the pedodontist and orthodontist, a good understanding of smile designing principles, and strict adherence to bonding procedures are key factors for treatment success.

## Figures and Tables

**Figure 1 children-10-00546-f001:**
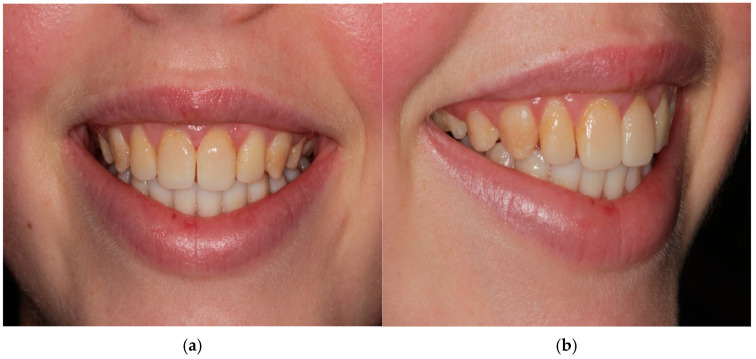
(**a**,**b**): Case 1—preoperative smile photographs.

**Figure 2 children-10-00546-f002:**
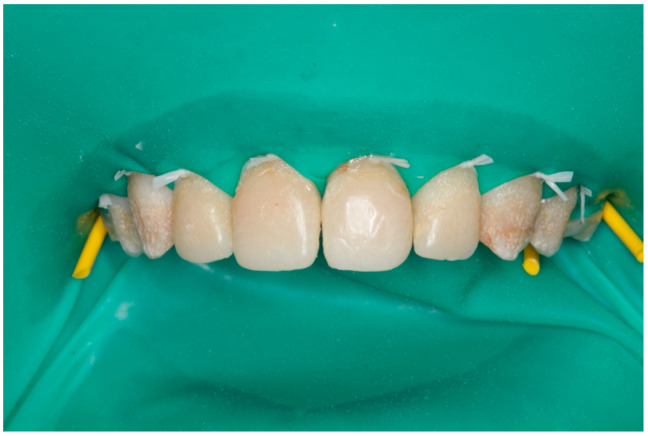
Case 1 clinical procedure.

**Figure 3 children-10-00546-f003:**
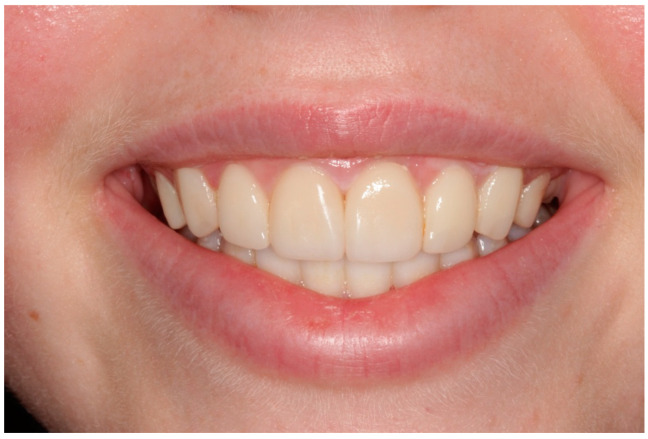
Case 1 postoperative smile photograph.

**Figure 4 children-10-00546-f004:**
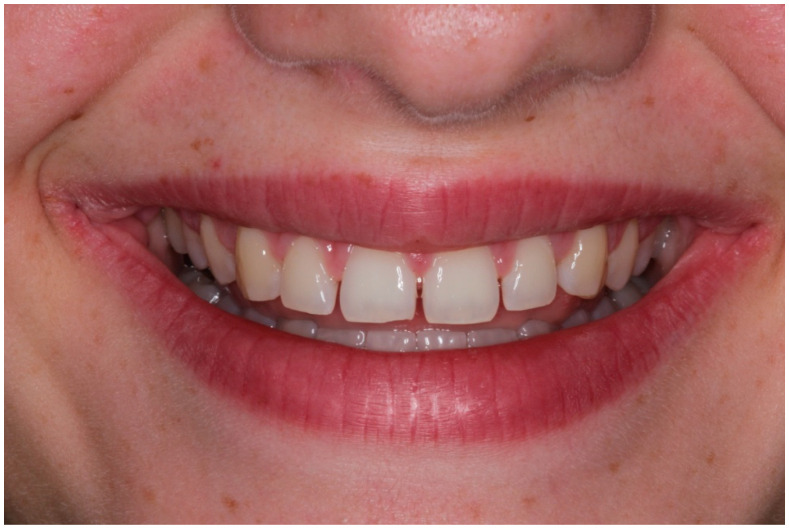
Case 2 preoperative photograph.

**Figure 5 children-10-00546-f005:**
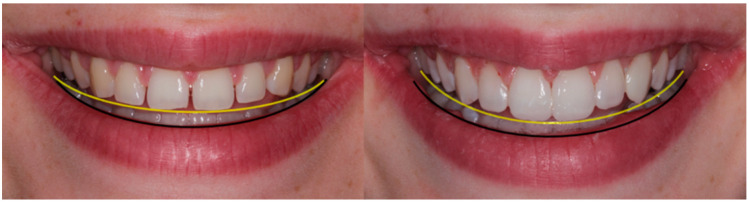
Case 2 smile designing considering incisal edge parallelism to lower lip contour.

**Figure 6 children-10-00546-f006:**
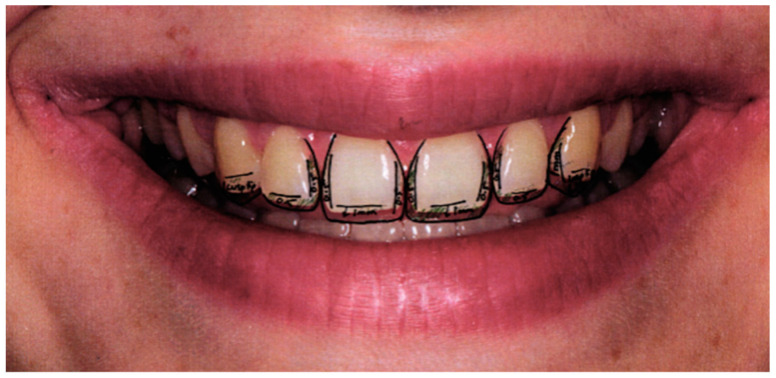
Case 2 smile designing and communication with the orthodontist.

**Figure 7 children-10-00546-f007:**
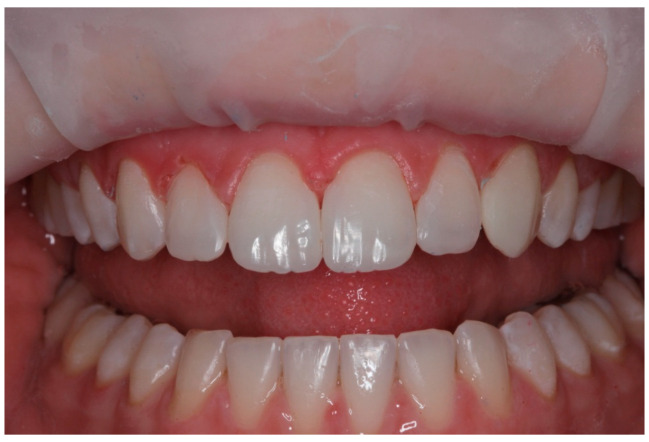
Case 2 post-operative photograph showing surface details with macro- and micro-anatomy.

**Figure 8 children-10-00546-f008:**
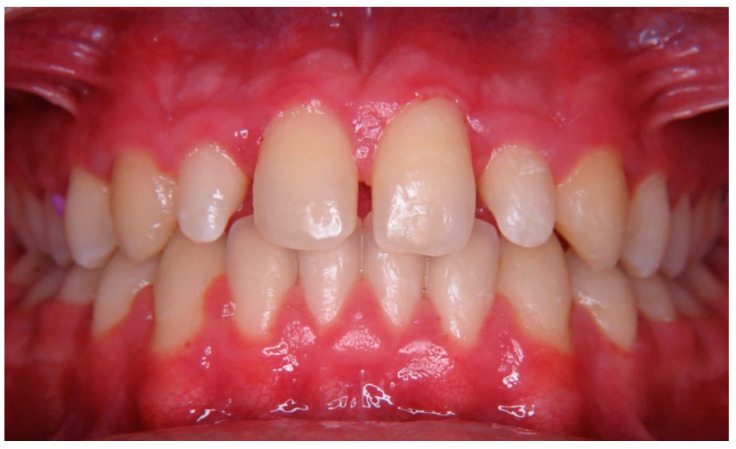
Case 3 preoperative photograph.

**Figure 9 children-10-00546-f009:**
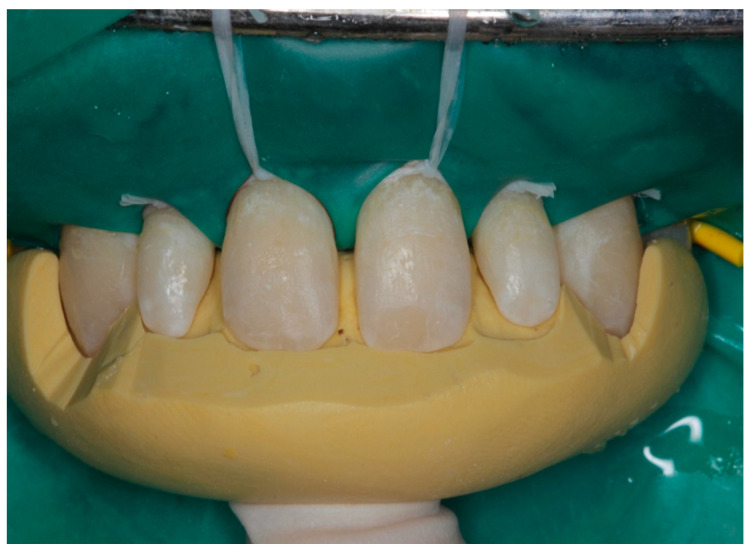
Case 3 clinical procedure.

**Figure 10 children-10-00546-f010:**
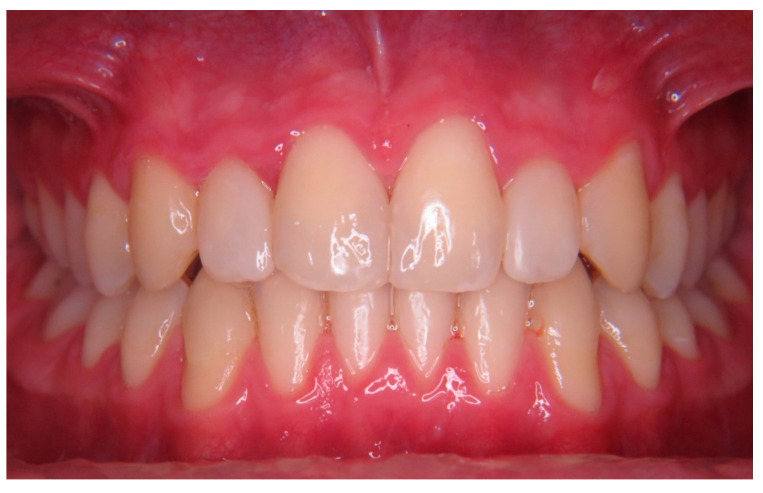
Case 3 postoperative.

## Data Availability

All the data regarding the study are with the main author and will be shared when required.

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
