# Peer review of "Esthetic Rehabilitation of Pediatric Patients Using Direct Bonding Technique—A Case Series Report"

_children, 2023, doi:10.3390/children10030546_

Round 1

Reviewer 1 Report

In this case series, three different pediatric patients with varying aesthetic complaints were selected. The article reported the variety of esthetic treatments emphasizing the most conservative treatment modality for anterior aesthetic rehabilitation by using direct composite veneer restorations. 

It is relevant as a case series that helps clinicians with one of the treatment modalities. 

To a certain extent as the veeners are not commonly used in pediatric patients. It is concerned the cases presented in the manuscript. 

The references are appropriate. 

Pertaining to figures- The age of the patients should be cross-checked as the patients look older than the age reported in the manuscript. 

Well-written manuscript. I would like the authors to check the age of the third case presented in the manuscript. Add the recent reference article if possible

Author Response

We thank you for allowing us to revise and resubmit with the suggested modifications by the reviewers. We like to thank the reviewers for their valuable comments that helped us to re-edit the manuscript for better clarity. All the modified areas are highlighted in blue within the manuscript.

Reviewer 1

In this case series, three different pediatric patients with varying aesthetic complaints were selected. The article reported the variety of esthetic treatments emphasizing the most conservative treatment modality for anterior aesthetic rehabilitation by using direct composite veneer restorations. It is relevant as a case series that helps clinicians with one of the treatment modalities. To a certain extent as the veneers are not commonly used in pediatric patients. It is concerned with the cases presented in the manuscript. 

 Thank you for your valuable suggestions. We have added 5 more references in the discussion citing the use of composite veneers in pediatric cases. All changes have been highlighted in blue and track changes have been activated.

Composite veneers are recommended as the most conservative treatment option in pediatric patients with discolored, traumatic, or teeth with hypoplastic changes. The direct bonding technique has many advantages compared to the few disadvantages such as the need for proper isolation, skill, and additional chairside time [16]

The longevity of composite as an anterior restorative material was evaluated by Comba et al in a five-year retrospective study on post-orthodontic patients. They concluded that even after 5 years composite when used as an anterior esthetic corrective material had satisfactory results.  The few cases that failed were repairable and hence composites are the most conservative and ideal treatment for teenagers and growing children [17] The advantage of using the direct bonding technique is that it can be polished and finished adequately to prevent any gingival irritation. The contraindications mentioned in the literature include cases where isolation cannot be achieved, caries-prone teeth, and lack of patient follow-up[18] Recen et al compared 15 cases each done by indirect composite and direct composite techniques. They concluded that the direct technique is significantly better than the indirect composite veneer technique. According to their findings, the indirect composite veneers were more prone to discoloration and required complex preparation such as impression technique, and lab procedures.[19] In another study, the type of composite used for the direct technique was evaluated by Mewan, he concluded that the nanohybrid composite used with an Indirect –Direct technique had the maximum longevity compared to micro-filled composites. In both direct and indirect techniques the nanohybrid composites had decreased gingival microleakage than the micro-filled composites[20]

[16]      W. W. Johnson, “Use of laminate veneers in pediatric dentistry: present status and future developments.,” Pediatr. Dent., vol. 4, no. 1, pp. 32–37, Mar. 1982.

[17]      A. Comba et al., “5-year retrospective evaluation of direct composite restorations in orthodontically treated patients,” J. Dent., vol. 104, Jan. 2021, doi: 10.1016/j.jdent.2020.103510.

[18]      K. J. Donly and F. García-Godoy, “The use of resin-based composite in children,” Pediatr. Dent., vol. 24, no. 5, pp. 480–488, Sep. 2002.

[19]      D. Recen, B. Önal, and L. sebne. Turkun, “Clinical evaluation of direct and indirect resin composite veneer restorations: 1 year report,” J. Ege Univ. Sch. Dent., vol. 40, no. 2, pp. 103–115, 2019, doi: 10.5505/eudfd.2019.66933.

[20]      M. S. Abdulrahman, “Evaluation of the Sealing Ability of Direct versus Direct-Indirect Veneer Techniques: An In Vitro Study,” Biomed Res. Int., vol. 2021, 2021, doi: 10.1155/2021/1118728.

The references are appropriate. 

 Thank you

Pertaining to figures- The age of the patients should be cross-checked as the patients look older than the age reported in the manuscript. 

The third patient was born on 29/5/2002. At age of 12, his dentist sent his OPG with a referral letter to our orthodontist:

He was booked for an orthodontic assessment complaining of crossbite when he was 13 years old in 28/1/2015. He started fixed orthodontic treatment soon after:

At 16 after the orthodontic correction, the esthetic correction was done followed by regular follow-up annually

Reviewer 2 Report

Dear authors,
congratulations for the interesting and current topic addressed. To clarify some aspects, please consider the following:
caz1-the teeth that were reconstructed, were they endodontically treated?
case 2-did direct veneering cause changes in dynamic occlusion?
case 3-did the closing of the diastema cause periodontal changes?

Author Response

We thank you for allowing us to revise and resubmit with the suggested modifications by the reviewers. We like to thank the reviewers for their valuable comments that helped us to re-edit the manuscript for better clarity. All the modified areas are highlighted in blue within the manuscript.

 Reviewer 2

Well-written manuscript. I would like the authors to check the age of the third case presented in the manuscript. Add the recent reference article if possible.

The third patient was born on 29/5/2002. At age of 12, his dentist sent his OPG with a referral letter to our orthodontist:

He was booked for an orthodontic assessment complaining of a crossbite when he was 13 years old in 28/1/2015. He started fixed orthodontic treatment soon after:

At 16 after the orthodontic correction, the esthetic correction was done followed by regular follow-up annually

Dear authors,
congratulations on the interesting and current topic addressed. To clarify some aspects, please consider the following:

Thank you for your suggestions and positive comments.

caz1-the teeth that were reconstructed, were they endodontically treated?

None of the teeth were endodontically treated.

case 2-did direct veneering cause changes in dynamic occlusion?

No changes were observed in dynamic occlusion as this was evaluated on the day of restoration and all follow-up evaluations.

case 3-did the closing of the diastema cause periodontal changes?

The patients were followed up for three years with six-month recalls and no periodontal changes were observed.

Graphical Abstract:
